# The Immune System Response to 15-kDa Barley Protein: A Mouse Model Study

**DOI:** 10.3390/nu14204371

**Published:** 2022-10-18

**Authors:** Barbara Wróblewska, Ewa Kubicka, Ewelina Semenowicz, Anna Ogrodowczyk, Anita Mikołajczyk, Dagmara Złotkowska

**Affiliations:** 1Department of Food Immunology and Microbiology, Institute of Animal Reproduction and Food Research, Polish Academy of Sciences, Tuwima Str. 10, 10-748 Olsztyn, Poland; 2Department of Public Health, Faculty of Health Sciences, Collegium Medicum, University of Warmia and Mazury, 10-082 Olsztyn, Poland

**Keywords:** barley, food allergens, mucosal immunity, proliferation

## Abstract

Barley (*Hordeum vulgare* L.) proteins are taxonomically homologous to wheat proteins and react with sera from patients with baker’s asthma. In the current work, the crude extract of barley proteins was divided into six fractions on DEAE-Sepharose. Their immunoreactivity in reacting with sera from patients with a confirmed food allergy varied, and the 15-kDa fraction (B–FrVI) showed the strongest response. In silico analysis confirmed that 15-kDa B-FrVI protein belongs to the trypsin/amylase inhibitor family and to a group of MHC type II allergens. In the next step, the immunogenicity of the B-FrVI was examined in a mouse model. It was shown that, compared to the PBS group, administration of B-FrVI to mice induced almost 2× higher amounts of specific IgG, ~2^17^, and IgA ~2^9^, as early as day 28 after immunization, regardless of the route (intraperitoneal or oral) of antigen administration (*p* < 0.0001). An ELISpot for B-cell responses confirmed it. Stimulation of mesenteric lymphocytes with pure B-FrVI significantly increased (*p* < 0.001) the proliferation of lymphocytes from all groups compared to cells growing in media only and stimulated with lyophilized beer. The experiments prove the strong immunogenicity of the 15-kDa B-FrVI protein and provide a basis for future studies of the allergenic nature of this protein.

## 1. Introduction

Barley is a rich source of carbohydrates, proteins, lipids, vitamins, and minerals [1]. Carbohydrates—the main chemical component—account for 80% of the barley grain and are known for their unique composition, as 4–9% of them is *β*-glucan, which plays an essential role in the reduction of cholesterol and serum glucose [1,2]. Whole grain consumption (such as natural pearl) is a helpful diet element in reducing the risk of cardiovascular disease, colonic inflammation, hypertension, and obesity [3,4]. Thanks to these properties, barley is beginning to be seen as a cereal that can produce healthy and functional foods. Researchers have reviewed the physicochemical properties of barley, but still, there is an insufficient number of studies concerning the immunogenic and antigenic properties of the proteins of this cereal.

The complexity of food matrices, the growth of dietary supplementation, and new food trends have prompted detailed studies of individual food components. From 2019–2021, the list of allergens compiled by the WHO/IUIS Allergen Nomenclature Database (http://allergen.org) (accessed on 1 July 2022) increased by 106, including food allergens and allergens of other origins (inhalant, pollen), which also pose a risk through cross-reactivity [5]. Food allergy is a common problem, and it is assessed that up to 10% of adults can be affected [6]. Barley is closely taxonomically related to wheat [7], which has 124 proteins and isoforms described as allergens (https://www.allergome.org/script/search_step2.php, accessed on 2 February 2022). Furthermore, a case study reported on barley allergy and presented data from adult patients with diagnosed baker’s asthma [8]. Hence, there is a need to research barley proteins’ immunogenicity.

Barley grains contain 10–20% of proteins which are accumulated in their endosperm. Depending on their solubility, they represent albumin (water-soluble), globulin (salt-soluble), prolamin and hordein (alcohol-soluble), and glutelin (alkali-soluble) [9]. After digestion, they are a source of peptides which induce a cascade of immune system responses. In the case of gluten fractions, this is critical and toxic for celiac patients [10]. Barber et al. (1989) [11] presented the immune potential of the salt-soluble barley protein, which represents the family of α-amylase/trypsin inhibitors. The authors used sera of patients with diagnosed baker’s asthma disease to determine the immune reactivity of crop proteins. The band with a molecular weight of about 14.5 kDa showed strong IgE reactivity in the RAST inhibition assay. Wheat proteins are prime allergens inducing baker’s asthma. Barber’s [11] experiments proved a high homology between barley and wheat IgE epitopes, which conduct cross-reaction and disease symptoms. Armenita et al. (1993) [12] showed that a purified mixture of barley α-amylase and trypsin inhibitors induces a strong response in skin prick tests in allergic patients. Garcia-Casado et al. (2001) [13] presented a dot blot-positive reaction to the lipid transfer protein 1 (9 kDa, LTP1) and Z4 protein (45 kDa) from barley, with four beer-allergic patients’ sera. The work of Lee et al. (2020) [14], presented barley as a significant allergen in a population of Korean children. The barley-allergic group was characterized by food allergies other than barley, including as many as 75% who were allergic to wheat. In this group, the authors observed an anaphylactic reaction after barley injection in seven of the twenty subjects.

Therefore, the present study aimed to investigate the immunogenic potential of a selected IgE-reactive barley fraction protein with a low molecular mass. Humoral and cellular responses were monitored in experiments to test immunoreactivity and immunogenicity.

## 2. Materials and Methods

### 2.1. Raw Barley Grains

The investigated material was the dehulled seeds of a Polish cultivar of spring barley (*Hordeum vulgare* L.) Poldek (POB 2295). The grains were obtained from Olsztyn Seed Central OLZNAS—CN (Olsztyn, Poland).

All chemicals used in these studies were bought from Sigma-Aldrich (Poznań, Poland).

### 2.2. Beer

We bought the beer at the local market. According to the manufacturer, the beer was of pilsner type, light, 12.5% extract (w), and 5.2% alcohol (v). After lyophilization, a solution at a 100 μg/mL concentration was added to the mice lymphocyte cultures as a stimulation agent.

### 2.3. Protein Extraction and Purification

The barley seeds were ground using a laboratory mill (Laboratory Mill 3303; Perten Instruments, Segeltorp, Sweden), and the proteins were extracted with 0.025 M of phosphate buffer, pH 8.0, at a ratio of 1:3 (*w*/*v*). The extraction was performed for 1 h, mixing every 10 min at temperatures 0–4 °C. The mixture was centrifuged on an Eppendorf 5418R centrifuge (Eppendorf, Hamburg, Germany) at 3500× *g* for 20 min. The pellet was discarded, and the supernatant (crude barley protein extract, cBP) was subjected to further purification and analyses.

#### Protein Purification

The crude protein of barley extract (cBP) (4.8 mg/mL), diluted in buffer A (25 mM phosphate buffer, pH 8.0), was loaded on a column of DEAE Sepharose Fast Flow (1.6 cm × 85 cm; Pharmacia LKB, Uppsala, Sweden). Proteins were eluted in 7-mL fractions at a 1.5 mL/min flow rate in a gradient of 0–100% of buffer A containing 0.5 M NaCl. The elution profile was monitored at 280 nm using a UV-1800 Shimadzu spectrophotometer (Shimadzu Corporation, Kyoto, Japan). The eluent from the subsequent tubes corresponding to the protein peaks was pooled, lyophilized, and referred to as the protein fractions.

### 2.4. Determination of Protein Concentration

The total protein concentration in the samples was determined using Bradford’s Reagent (B6916; Sigma-Aldrich). Bovine serum albumin (BSA) was used for the standard curve with a concentration range of 0–1.4 mg/mL. Absorbance was read at *λ* = 595 nm on a Jupiter UVM-340 spectrophotometer (ASYS Hitech GmbH, Eugendorf, Austria). The final results were a mean of three replicates ± SD.

### 2.5. Sodium Dodecyl Sulfate−Polyacrylamide Gel Electrophoresis (SDS-PAGE)

The crude barley extract (cBP, at a concentration of 1 mg/mL) and its fractions eluted by DEAE ion-exchange chromatography (15 μL/per lane) were subjected to SDS- PAGE according to the standard protocol used in the laboratory [15]. Briefly, the proteins were separated on gels (15% for cBP and 12.5% for fractions), under a constant current of 30 mA, using a Mini-PROTEAN system (Bio-Rad Laboratories, Inc., Warsaw, Poland). The gels were stained with 0.1% Coomassie Brilliant Blue R-250 dye (Sigma-Aldrich, Poznań, Poland). We used Sigma Markers in the range of 6.5 - 66 kDa (Sigma-Aldrich, Poznań, Poland) to estimate the proteins’ molecular weight profile in cBP and barley protein fractions.

### 2.6. Identification of IgE Reactive Barley Fraction

The human sera used in the study were collected during the previous project, “The Influence of Fermented Cow’s Milk Products Displaying Reduced Antigenicity on the Immunological Response of Warmia and Mazury Region’s Patients with Food Allergy with Consideration of Genetic Aspects” (No. N312 311939). All procedures were approved by the local ethical committee (No. 2/2010). All sera were characterized with the Pediatric and Atopy Screen tests (EUROIMMUN AG, Lübeck, Germany). For the study, we chose 5 serum samples from patients aged 35–42, with total IgE >age norm and clinical allergic manifestations to various food-origin allergens (FA-PS), and 5 serum samples from patients aged 30–42, with total IgE <=age norm, from healthy people (nA-PS) [16]. Sera were pulled and stored at −20 °C until use.

#### Indirect IgE ELISA

With modifications, the ELISA was performed according to the protocol of Zhang et al. (2019) [17]. Briefly, 400 Nunlock microplates (Greiner Bio-One GmbH, Frickenhausen, Germany) were coated with antigen (cBP, barley protein fractions) (50 μg/mL). After the incubation, blocking, and washing steps, the human sera (FA-PS or nA-PS) were added to the plate for overnight incubation with the anti-human IgE biotin-labeled antibodies (ab99807; Abcam, Cambridge, UK) for 1.5 h at 37 °C. After washing, the plate was incubated with streptavidin-HRP (ab 7403; Abcam, Cambridge, UK) for 0.5 h at 37 °C. After the final wash step, the TMB substrate (Abcam, Cambridge, UK) was added to develop a color reaction which was stopped by adding 50 μL H_2_SO_4_ (2M). The optical density (OD) was measured at 405 nm using the Jupiter UVM−340 spectrophotometer (ASYS Hitech GmbH, Eugendorf, Austria). Each barley fraction was subject to reaction with the FA-PS and nA-PS sera as a negative control in triplicates. A mean OD value ± SD was considered positive if it exceeded the mean of the negative controls by more than three standard deviations.

### 2.7. Biological Experiment

Female Balb/C mice (6–8 weeks old) were purchased from the Medical Research Center, Polish Academy of Sciences (Warsaw, Poland). The mice were housed in the animal facility of the Institute of Animal Reproduction and Food Research, Polish Academy of Sciences (Olsztyn, Poland) in individually ventilated cages. The mice had ad libitum access to a total pathogen-free modified, gluten-free diet (1434; Altromin Spezialfutter GmbH, Lage, Germany) [18] and sterile drinking water. The Local Care Use of Animals Committee approved all procedures performed with experimental animals in this study (Permit No. 43/2015/N).

During the experiment, the mice were randomly divided into three groups (six mice each), two treated with the barley fraction VI (B–FrVI) and a control group treated with PBS (10 mM phosphate buffer, pH = 7.4) according to the scheme presented in Figure 1.

The groups of mice were treated as follows:

Group ip.B−FrVI: One day before the experiment started (−1 d), the mice were injected intraperitoneally (ip.), with a mixture of 100 μg of B−FrVI with Freud’s Adjuvant, in a total volume of 200 μL. On days 0, 7, and 14, the mice were gavaged (o.) with 1 mg of B−FrVI with the cholera toxin (CT) as an adjuvant, in a total volume of 200 μL. 

Group o.B−FrVI: One day before the experiment started (−1 d), the mice were intraperitoneally (ip.) injected with 200 μL of PBS. On days 0, 7, and 14, the mice were gavaged (o.) with 1 mg of B−FrVI with the CT as an adjuvant, in a total volume of 200 μL.

Group PBS: One day before the experiment started (−1 d), the mice were intraperitoneally (ip.) injected with 200 μL of PBS. Then, on days 0, 7, and 14, the mice were orally administered with 200 μL of PBS.

The 20-ga gavage needle (0.9 mm OD × 0.6 mm ID × 38 mm long) was attached to a 1-mL tuberculin syringe for the oral agent administration. Intraperitoneal immunization was performed using a tuberculin syringe with an 18-ga needle (0.45 mm OD × 16 mm long).

On day 14 after primary exposure, blood and fecal samples were collected weekly from individual mice to measure specific B−FrVI immunoglobulin class G and A titers. On day 48, the mice were euthanized by carbon dioxide inhalation, and single-cell suspensions were prepared from their spleens (SPL) and mesenteric lymph node (MLN) lymphocytes according to the standard protocol used in the laboratory [19].

#### Sample Collection

Serum samples from the mice were collected by puncture of the submandibular vein. The blood was allowed to stand for 1 h at room temperature, then centrifuged using the Eppendorf 5418R centrifuge (Eppendorf, Hamburg, Germany) at 12,000× *g* for 10 min at 4 °C to separate the serum. Then, sera were collected and stored at −20 °C for further analysis.

Freshly collected fecal pellets were dissolved in 0.1 M PBS, containing 0.1% NaN3 (pH 7.2), in a *w*/*v* ratio of 100:1. After homogenization using Fugamix (ELMI LTD, Riga, Latvia), samples were centrifuged on the Eppendorf 5418R centrifuge at 12,000× *g* for 10 min. Supernatants were collected and stored at −20 °C for further analysis [20].

### 2.8. Mouse Humoral Response Determination

#### Indirect ELISA

Specific to barley, B−FrVI antibodies were titered in serum and fecal extracts were measured by indirect ELISA. The 400 Nunlock microplates (Greiner Bio-One GmbH, Frickenhausen, Germany) were coated with B−FrVI (5 µg/well in PBS) and incubated at 37 °C for 1 h. Next, the plates were blocked with 1.5% gelatin diluted in PBS and incubated under the same conditions. Following blocking, the microplates were washed with PBST (PBS + 0.05% Tween 20). Serial dilutions of the serum and feces extract (50 µL) were added to the plates and incubated for 1 h at 37 °C. After a washing step, the plates were incubated with HRP-labeled secondary antibodies for 1 h at 37 °C. After washing, the ABTS reagent (Roche Diagnostics GmbH, Mannheim, Germany) was added. The optical density (OD) was measured at 405 nm on the Jupiter UVM-340 spectrophotometer (ASYS Hitech GmbH, Eugendorf, Austria). The endpoint titers (EPT) were expressed as the reciprocal dilution of the last sample dilution of 0.1 OD above the negative control [20].

### 2.9. Lymphocyte Isolation

The isolated SPL or MLNs were homogenized in a glass homogenizer containing 3 mL of incomplete medium (IM, RPMI-1640 with L-glutamine supplemented with 1 mM of HEPES and 10 units/mL of penicillin-streptomycin solution). The cell suspensions were filtered through an 80-μm mesh membrane and centrifuged at 400× *g* at 10 °C for 10 min on the Eppendorf 5418R centrifuge. Cell pellets from the SPL were additionally treated with RBC lysis buffer (R7757; Sigma-Aldrich, St. Louis, MO, USA) [21] to remove the red cells. After final washing, the cell pellets were resuspended in 1 mL of IM and stained with trypan blue to adjust the cell number in each sample.

### 2.10. Lymphocyte Culture

Mononuclear SPL and MLN cell suspensions were plated on 96-well culture plates with 10^5^ cells per 200 μL. Cells were cultured in a complete medium (CM, RPMI 1640 (Sigma-Aldrich) supplemented with 10% heat-inactivated fetal bovine serum, 1 mM of nonessential amino acids (Sigma-Aldrich), and 10 U/mL of penicillin-streptomycin solution (Sigma-Aldrich)). The cultures were stimulated with B–FrVI, or beer lyophilize at a concentration of 100 μg of protein/mL, at 37 °C in a 5% CO_2_ atmosphere for 96 h. Cells growing in a medium, only, served as a control.

#### 2.10.1. Proliferation Assay

To check the cell’s proliferation capability, we used the EZ4U ELISA Assay Kit (Biomedica Medizinprodukte GmbH & Co KG, Vienna, Austria). The kit is a nonradioactive proliferation assay that converts tetrazolium salt to formazan, resulting in a color change. We followed the manufacturer’s protocol. After 72 h of splenocyte and MLN cell culture, 25 μL of substrate were added to each well. After 4 h of incubation, the optical density was read for each well at *λ* = 450 nm using the Jupiter UVM-340 spectrophotometer (ASYS Hitech GmbH, Eugendorf, Austria). Results were presented as the mean ± SD from five replicates [20].

#### 2.10.2. B-Cell ELISpot

The ELISpot, an assay for measuring the total immunoglobulin and antigen-specific antibody-producing cells (APC), was conducted according to Chudzik-Kozłowska et al. (2020) [15]. A mixed cellulose ester membrane-bottomed microtiter plate (Millipore) was coated with 50 μg/mL of antigen or non-labeled rat anti-mouse IgG (M5899; Sigma-Aldrich), IgA, (A4789; Sigma-Aldrich), and antibodies in sterile PBS, overnight at 4 °C. The wells were blocked with the CM for 1 h at 37 °C and washed four times with PBST. As a next step, 100 μL of varying concentrations of cell suspension were added to the wells, and the plate was incubated overnight at 37 °C in a 5 % CO_2_ atmosphere. After the washing step, anti-mouse IgG or IgA HRP conjugates were added. Then, the plates were set in a humidity chamber overnight at 4 °C. After washing, the color reaction was developed with TMB–H peroxidase substrate (TMBH-1000; Moss, Inc.) for ∼1 h. After additional washing with tap water, the APC were counted under a stereo zoom microscope (OLYMPUS SZX9). The results were expressed as the mean APC number from the three wells per 10^6^ cells ± SD.

### 2.11. In Silico Analysis of Immunoreactivity

Sequences of Hordeum vulgare (barley) proteins with molecular ranges of 8–24 kDa, were retrieved from the UniProt and NCBI databases in FASTA format, and used for the in silico analyses described by Ogrodowczyk et al. (2021) [22]. Briefly, proteins and fragments were selected following two criteria: (1) sequences deposited in allergen databases (Allergome, AllergenOnline V.21 and WHO/IUIS Allergen Nomenclature) https://www.allergome.org (accessed on 1 July 2022), http://www.allergenonline.org (accessed on 1 July 2022), and http://www.allergen.org (accessed on 1 July 2022); and (2) sequences not deposited in allergen databases but with reported sensitization potential. The mentioned protocols were based on: (1) full-length alignment against a database of known allergen sequences; (2) 80-mer amino acid alignment; and (3) amino acid exact matches of 8-mer (algorithms of www.allergenonline.org tool; accessed on 12 July 2022) and 6-mer (http://allermatch.org; accessed on 6 July 2022). The IgE epitope prediction was carried out using the AlgPred server (http://crdd.osdd.net/raghava/algpred; accessed on 14 July 2022)). Selected proteins were tested for binding to the human major histocompatibility complex class II (MHC II) with data available from the EpiTOP3 server (http://www.ddg-pharmfac.net/EpiTOP3/; accessed on 12 July 2022) [23]. EpiTOP3 is a server for human leukocyte antigen (HLA) class II binding prediction using proteochemometric models. This tool analyzes the potential binders to the most common alleles among the human population (12 HLA-DRB1, 5 HLA-DQ, and 7 HLA-DP). The threshold value of 6.3 was adopted to analyze peptides with IC50 < 500 nM, which were considered strong binders for MHC II. Peptides with IC50 < 500 nM (6.3 > pIC50 > 5.3) were considered as weak binders. Since the secretion of interleukin-4 (IL-4) is a characteristic of the T-helper 2 response, but MHC II epitope binding does not always induce IL-4, we used the IL-4 pre-tool (http://crdd.osdd.net/raghava/il4pred/; accessed on 25 July 2022) in the next step as a method to distinguish the peptide potential for the induction secretion of IL-4. For the same reason, we used an online module for screening peptides inducing the IFN-γ (https://webs.iiitd.edu.in/raghava/ifnepitope/; accessed on 25 July 2022). Default settings were used in the mentioned tools with an SVM threshold of 0.2 and overlapping peptides/epitopes with a window length of 9. As the peptide-binding groove of MHC class II proteins binds peptides between 3 and 15 amino acid residues or longer, we adopted the abovementioned criteria.

### 2.12. Statistical Analysis

The results were presented as the mean from each group ± standard deviation (SD). Statistical differences between experimental groups were evaluated using ANOVA tests followed by Tukey’s post hoc test. A *p*-value below 0.05 was considered significant. Analysis was performed with GraphPad Prism version 8.0.0 (GraphPad Software Inc., San Diego, CA, USA).

## 3. Results and Discussion

### 3.1. Characteristic Barley Protein

An awareness of healthy eating has prompted the search for raw food materials rich in nutrients and bioactive ingredients. They should also have technological properties suitable for food processing. Some of them, such as barley, are undervalued, but are being rediscovered. Its role in brewing has been known for centuries, but other bioactive compounds make it interesting as an additive for personalized food products. Case studies have presented allergic reactions after eating food containing barley proteins or drinking beer. Several peptides were found in barley and characterized as sensitizing the host by different routes of exposure (http://www.allergome.org/script/dettaglio.php?id_molecule=2040, accessed on 25 July 2022). Until today, the Comprehensive Protein Allergen Resource (http://db.comparedatabase.org, accessed on 25 July 2022) consists of 11 barley peptides with proven binding to IgE. The WHO/IUIS Allergen database includes six peptides or proteins qualified as food allergens and one air allergen, all of barley origin. However, research studies have still only recognized them as allergens based on the sera of a patient with baker’s asthma or as a food allergy concern of cereals. The estimated prevalence of cereal allergy in the EU is 1–2% depending on the geographical area, but it has a high economic impact [24]. Developing the food industry and a healthy modern diet increases the number of products consisting of wheat proteins. Antibodies induced by wheat proteins cross-react with taxonomically related ones from other species such as rye or barley. The data describing the allergenicity and immunoreactivity of particular barley proteins are still in progress, and using sera from patients with baker’s asthma [11,25] does not give complete information about the immune response directly to barley proteins.

In the presented study, we extracted proteins from barley var. Poldek seeds (cBP) with phosphate buffer (p. 2.3). The cBP, with a protein concentration of 4.8 mg/mL, was subject to 1-D SDS-PAGE.

We found that the protein pattern of the raw extract was characterized by the number of bands corresponding to proteins with a molecular weight (MW) below 66 kDa (Figure 2a). The bands with an intense color, visible on the electrophoretic gel, corresponded to proteins with an MW of about 56, 36, 32, 29, and 16 kDa (Figure 2a). The obtained barley protein pattern was similar to what Chmelik et al. (2002) [26] presented in water and salt extracts. The authors extracted proteins from milled barley seeds in four steps: the first extraction was with water, then the insoluble part was treated with NaCl solution, ethanol, followed by sodium hydroxide. Then, they identified the SDS-PAGE bands by MALDI-TOF [26]. They recognized β-amylase and α-amylase in the water and salt extracts. Our results showed that the one-step isolation of proteins with phosphate buffer is more effective than the two-step isolation. The bands corresponding to a weight below 16 kDa indicated a group of albumins and globulins, which are essential in the brewery industry because of their foam stabilization properties [27]. 

We tested the cBP extract with the sera of patients with confirmed food allergies using IgE ELISA. We found that the OD of allergic patients’ sera (FA-PS) was about three times higher than those of non-allergic patients (nA-PS). The assay confirmed the immunogenic potential of cBP and the presence of IgE immunoreactive proteins in the Poldek barley variety extract (Figure 2b). Using a bioinformatics tool, we performed the in silico analysis of barley proteins and peptides in the 8–24 kDa mass range (i.e., considered low molecular weight proteins) to predict the IgE occurrence of epitopes in the barley albumin (the results of the analysis are presented in Appendix A). We demonstrated the presence of 23 proteins or peptides with potential IgE binding sites, i.e., Hor v 20, Hor v 15, 16, or 17. Most of them have an established barley allergen status (WHO Allergen Nomenclature), according to the deep in silico analysis (this potential is hypothetical) or to the studies that used patients’ sera sensitive to wheat proteins (patients with baker’s asthma), which means that used antibodies cross-react with barley proteins. The safety of food products dedicated to customers suffering from barley allergy and celiac disease (CD) also enforces the validation of popular cereal products using barley malt, such as beer or bakery products. The growing trend of personalized diets requires a detailed characterization of proteins’ biological activity (including immunoreactivity, immunogenicity, and allergenicity), and their fractions in food.

The presented study separated the crude barley protein extract on the ion-exchange column according to the mentioned protocol (p. 2.3), yielding the elution profile illustrated in Figure 3a. We obtained six protein fractions: I–VI.

Each protein fraction was subject to SDS-PAGE analysis to evaluate the distribution of protein bands (Figure 3b). We found that the dominant proteins had a molecular weight between 36–45 kDa in fractions III–V, ~24 kDa in fractions III, IV, and a band with a weight between 14.4–20 kDa in fractions I, III, VI. As the next step, we examined the immunogenic potential of each fraction to bind the IgE from human serum (FA-PS and nA-PS) (Figure 3c). 

The fourth of the six tested fractions positively responded to the test. The highest affinity to IgE antibodies was presented by fraction VI (B-FrVI). This finding is in line with the discovery by Mena et al. [28], that a patient with baker’s asthma was sensitive to barley protein with a molecular weight of 14.5 kDa. This fraction was recognized as substantial in water–salt-soluble barley proteins. The authors characterized the barley endosperm protein as IgE-binding in profound molecular studies. They identified it as a glycosylated monomeric member of the multigene inhibitor family of α-amylase/trypsin from cereals. Based on all these findings, the B-FrVI fraction was selected for further mouse experiments as the one with the highest immunogenic potential in tests with patient sera (Figure 3c).

### 3.2. Immune Response to Barley

The ability of B-FrVI to induce an immune response was tested in a mouse experiment. The humoral response to B-FrVI was compared in mice by administering the antigen by oral gavage to the sensitized mice, and PBS to the not-sensitized control group (Figure 1).

We determined the specific IgG and IgA titers in the serum and fecal extracts during the experiment (Figure 4). We found a significant increase in the specific IgG (*p* < 0.001) and IgA (*p* < 0.001; *p* < 0.01, *p* < 0.05) titers in all of the collected sera in comparison to the control PBS group. After all treatments, on the 48th day of the experiment, the specific IgG titer reached the levels of 2^15.6 ± 0.55^ and 2^14.6 ± 2.5^ for the ip. and o.B-FrVI groups, respectively, compared to 2^5.75 ± 1.3^ for the PBS group (*p* < 0.1) (Figure 4a). Serum-specific IgA (Figure 4b) also had significantly higher titers when compared to the PBS group, with 2^6.2 ± 1.6^, 2^6.6 ± 0.5^, and 2^3.25 ± 0.5^, respectively, for the ip., o., and PBS. In the fecal extracts, we found anti-B-FrVIIgA at low levels. The EPT decreased during the time of the experiment. Despite this, on days 28 (2^3.2 ± 1.9^ and 2^4.4 ± 0.5^) and 35 (2^3.6 ± 1.6^ and 2^3.8 ± 0.8^), the EPTs of the coproantibodies were significantly higher (*p* < 0.5), than in the control PBS group (Figure 4c).

B-ELISpot analysis proved the increased activity of immune cells due to the administration of immunogenic barley fraction VI protein. We found that the initial sensitization of mice with B-FrVI (group ip.B-FrVI) did not increase the total IgG APC number compared to the orally gavaged group, o.B-FrVI (930 ± 18 and 1093 ± 15, respectively; Figure 5a). The specific IgG APC for ip.B-FrVI mice was significantly lower when compared to the o.B-FrVI and PBS groups, 573 ± 28, 660 ± 23, and 100 ± 13, respectively (*p* < 0.001) (Figure 5a). The opposite tendency showed IgA APC in splenocytes. Both the total and anti-B-FrVI APC were higher in ip.B-FrVI mice, at 33 ± 6 and 27 ± 1, when compared to the o.B-FrVI group, at 17 ± 6 and 13 ± 1, and PBS group (*p* < 0.001) (Figure 5b). All results differed significantly from the control PBS group (*p* < 0.001). The effects of the humoral and cellular response clearly showed the high immunogenic potential of B-FrVI to induce B-cells to produce specific antibodies and cause a cascade of immune reactions. The spleen is connected to the bloodstream, and all antigens enter it and activate different cells, which backed the B-cell follicles’ circulation [29]. Earlier studies [15] showed changes in the T-cell profile of sensitized splenocytes in response to stimulation with pea allergens, while mice showed no signs of allergy. Toyashima et al. (2017) found the spleen to be the main organ where the mast cells responsible for developing Th2-type food allergy are induced [30].

The similar humoral response in the two experimental groups (ip. and o. treated with antigen) let us hypothesize that protein B-FrVI passes through the intestinal mucosa and is processed in inductive lymphoid tissue, i.e., MLNs, to induce a systemic immune system response. The mechanisms by which B-FrVI protein induces and transmits the immune response require future studies at the molecular level.

Beer has been a known beverage for centuries, produced from barley malt. The SDS-PAGE of beer proteins showed protein fractions with molecular weights of about 55, 37, 14, and one below 10 kDa (Appendix A). The band with a MW ~14 kDa corresponded to B-FrVI. The BK samples were tested with FA- and nA-patient sera for the presence of IgE reactive proteins. The OD of the IgE in the reaction with FA-PS was 0.67 ± 0.03, which was assigned as a positive response compared to 0.34 ± 0.02 for nA-PS, (Appendix A). BK showed immunogenic potential, and we chose beer for stimulation in the ex vivo experiment. We set splenocytes and lymphocytes from MLN (Figure 6a,b) cultures. After 72 h of stimulation, the cultures were subject to a proliferation test (*p*. 2.10.1). Differences in OD levels indicated activation of the proliferation of immune cells by the proteins added to the media. Cells that grew in a medium served as a negative control.

We found that the cultures stimulated with purified B-FrVI significantly increased the OD value (*p* < 0.05) compared to the controls or group stimulated with lyophilized beer in both the splenocyte and MLN cultures. The B-FrVI increased the OD of the splenocyte cultures to 1.97 ± 0.06, 1.48 ± 0.12, and 0.87 ± 0.07 for the ip.B-FrVI, o.B-FrVI, and PBS groups, respectively (Figure 6a), compared to the controls, whose OD ranged from 0.5–0.25 for the respective group. The OD of the MLN cultures (Figure 6b) increased to about 0.8 after B-FrVI stimulation. It was about two to three times higher than the MLN cells grown in a medium only (*p* < 0.5). Stimulation of the MLN cells with BK did not exceed the OD level of the controls, except for the lymphocytes from o.B-FrVI, for which the OD was about two times higher than the controls, of 0.44 ± 0.07 vs. 0.23 ± 0.09, respectively (Figure 6b). Differences in the immunoreactivity of the BK and B-FrVI proteins suggest their different immunogenicity and allergenic potential.

The above experiment demonstrated the effect of B-FrVI on the proliferation of a mixture of immunocompetent cells isolated from effector (SPL) and inductive tissue (MLN). We used cells from sensitized and PBS mice. Sensitized cells intensively responded to the antigen in the medium culture and increased proliferation compared to cells growing in the medium. Such a reaction demonstrates the significant immunogenic potential of barley B-FrVI protein.

In conclusion, purified B-FrVI protein with a molecular weight of about 15 kDa exhibits immunogenic properties. It induces specific humoral and cellular responses and intense ex vivo proliferation of immune cells. In silico analysis revealed that the described barley protein, FrVI, is highly likely to induce an immune response via the MHC II pathway. Our findings highlight the B-FrVI protein is strongly immunogenic, and a positive reaction with the sera of food allergic patients indicates B-FrVI allergenic potential. Confirming the allergenic properties of B-FrVI protein requires detailed studies of this protein’s induction pathway of T and B cells. This is important because barley proteins are present in a wide range of food products.

## Figures and Tables

**Figure 1 nutrients-14-04371-f001:**
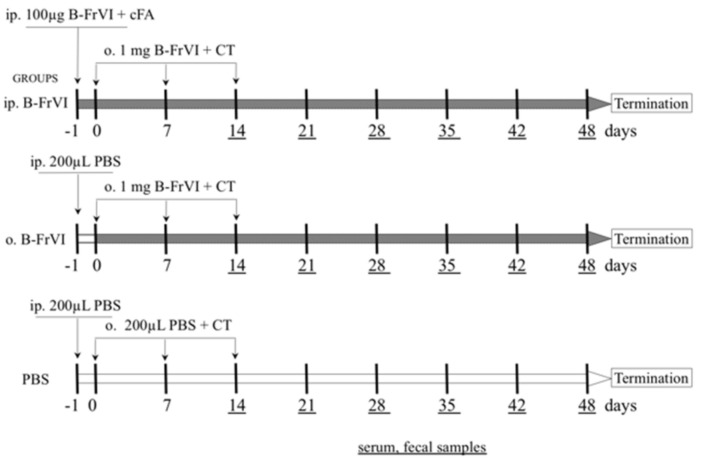
Animal experiment scheme. Abbreviation: o.− oral immunization; ip.− intraperitoneal immunization; B−FrVI− barley fraction VI proteins.

**Figure 2 nutrients-14-04371-f002:**
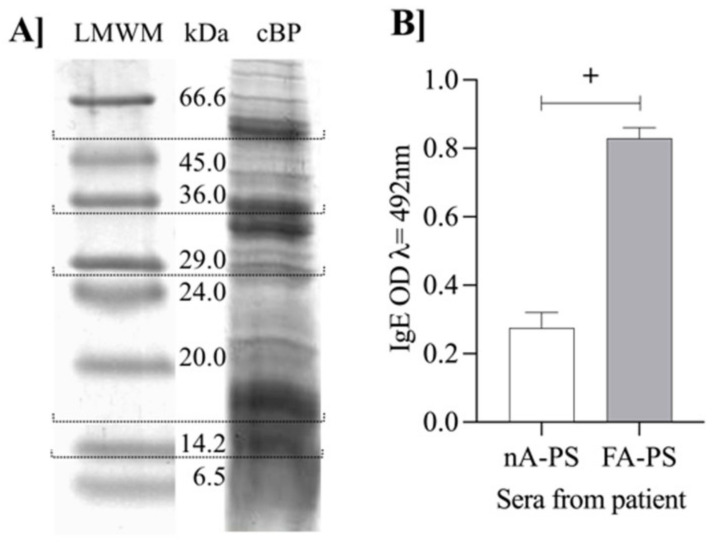
The barley crude extract (**A**) SDS-PAGE profile and (**B**) IgE immunoreactivity. Abbreviations: LMWM—low range molecular weight marker; cBP—crude extract of barley proteins; nA-PS—sera from patients non-allergic to food; FA-PS-sera from patients allergic to food. The mean OD value ± SD was considered positive (assigned with +) if it exceeded the mean of the negative controls (nA-PS) by more than three standard deviations.

**Figure 3 nutrients-14-04371-f003:**
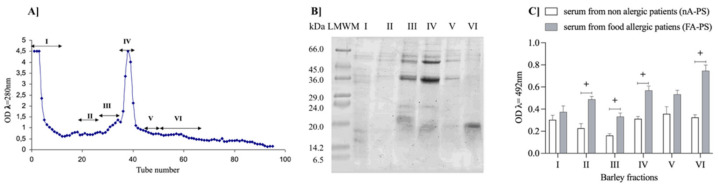
(**A**) Elution profiles of crude protein extract (cBP) from barley grains on the ion-exchange DEAE-Sepharose column; (**B**) SDS-PAGE protein patterns of eluted fractions I–VI (MWS, molecular weight standard); and (**C**) the IgE immunoreactivity of barley fractions I–VI. The OD value was considered a positive response with food-allergic patients’ serum (+) if it exceeded the mean of the controls (non-allergic patients’ serum) by more than three standard deviation values.

**Figure 4 nutrients-14-04371-f004:**
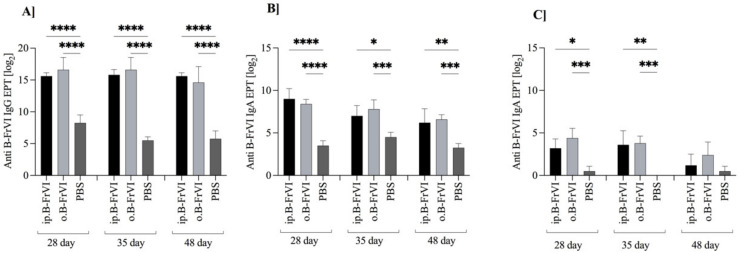
The humoral immune response of Balb/c mice to barley fraction VI (B–FrVI) to: (**A**) specific serum IgG; (**B**) specific serum IgA; and (**C**) specific IgA in fecal extracts. Mice were divided into three groups: sensitized (ip.B–FrVI, black bars), gavaged with antigen (o.B–FrVI, light grey bars), and PBS treated as control (PBS, dark grey bars). Data are expressed as the mean ± SD from the group (n = 6). Two-way ANOVA performed the statistical analysis. Statistical differences were assigned * for *p* < 0.05; ** for *p* < 0.01; *** for *p* < 0.001; and **** for *p* < 0.0001.

**Figure 5 nutrients-14-04371-f005:**
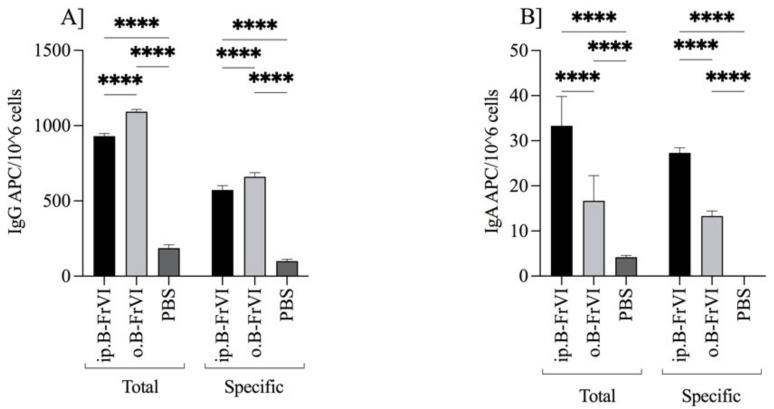
The (**A**) IgG and (**B**) IgA APC responses by Balb/c mice intraperitoneally (ip.B-FrVI) or orally (o.B-FrVI) sensitized with barley fraction VI, and by the control PBS-treated group (PBS). Splenocytes were isolated at the end of experiment from each group (n = 6) and set for B-cell ELISpot assay. The bars represent the mean of the group (n = 6) ± SD. Two-way ANOVA performed the statistical analysis. Statistical differences were assigned **** for *p* < 0.001.

**Figure 6 nutrients-14-04371-f006:**
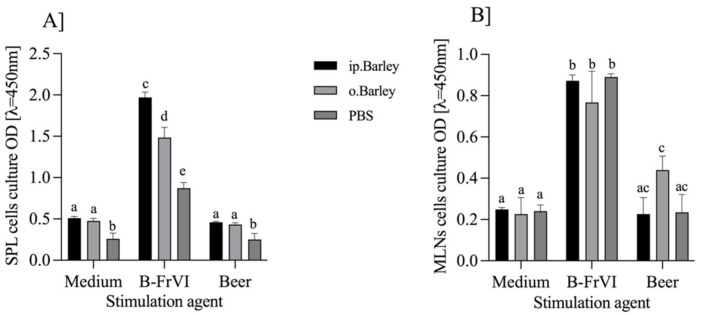
The (**A**) proliferation assays of splenocytes and (**B**) mesenteric lymph node lymphocytes after stimulation with B-FrVI, beer liophylisate (Beer), or growing in medium only (Con). Two-way ANOVA with Tukey’s test was used to determine the statistical differences. Bars represent the mean of the group (n = 5) ± SD. The mean values with different superscripts were different at *p* < 0.001.

## Data Availability

Not applicable.

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
