# Peer review of "The Immune System Response to 15-kDa Barley Protein: A Mouse Model Study"

_nutrients, 2022, doi:10.3390/nu14204371_

Round 1

Reviewer 1 Report

Review manuscript #: Response of the immune system to barley protein fraction of about 15kda - mice model study

Overall, the content of this manuscript is interesting and falls into the scope of the Nutrient journal. This reseach has some significant impacts can contribute to science. However, there are still several points which authors should improve before considereing for acceptance. My decision is major revision. Below are some raising points which I hope authors could consider for improving manuscript.

Tittle: need to think a new suitable tittle. The term “of about” is not suitable actually.

Abstract: The current abstract needs a major revision. It is not inclusive. Especially, what is the really finding of this research. The current conlcusion is still too general. L11-14: the first 3 sentences in abstract is too general information which is not needed to be included. Better remove.

Introduction

L27-37: these two paragraphs should be removed. The information is too general and not needed to state in the introduction.

Authors should also check again through the introduction part.

Material and methods: More information about the barley is expected

Figure and Tables: Some figures are too low resolution, please improve it.

Author Response

Dear Reviewer,

Thank you for your review and interest in the topic.

In the 21st century, there is an increase in the population of allergies, obesity problems, and metabolic disorders. The overlap of these diseases affects immune metabolism. Identifying and accurately characterizing primary food allergens will facilitate the modification of technological processes toward producing hypoallergenic proteins/foods and personalizing diets. The development of in silico methods facilitates research, but protein activity should be confirmed by in vitro experiments. Our goal was to predict the immunogenicity of protein fractions (in silico study) with various molecular weight ranges to determine whether all proteins induce the same strong reactions or are more immunogenic, which may translate into further technological activities. The 15kDa barley fraction showed immunogenic properties. As a fraction of nutritionally valuable grain, it should be further analyzed.

Thank you for your valuable comments. We have responded to them. We hope that the applied changes will find the acceptance.

Title: need to think a new suitable title. The term "of about" is not suitable, actually.

Abstract: The current abstract needs a major revision. It is not inclusive. Especially, what is the really finding of this research. The current conlcusion is still too general. L11-14: the first 3 sentences in abstract is too general information which is not needed to be included. Better remove.

The title and Abstract are, at the moment, the most important and, more difficult part of the research paper. We changed the title for:

The immune system response to 15kDa barley protein fraction - mice model study.

We rewrite the abstract for better characterization of our investigations. We hope the Reviewer will find it satisfactory.

Introduction

L27-37: these two paragraphs should be removed. The information is too general and does not need to state in the introduction.

Thank you for this comment. We agree, and we removed those two paragraphs.

The authors should also check again through the introduction part.

Thank you for this comment. We have rechecked the introduction. We corrected some errors and the reference numbering. The changes have been highlighted in red. We added an argument supporting the idea of the need to study both the immunogenicity and allergenicity of barley proteins. 

Material and methods: More information about barley is expected

We bought the seeds from certified Seed Central in Olsztyn. The seeds were certified for quality, but little information has been included. We added two annotations that the barley was dehulled (we missed it earlier), and Poldek is a spring cultivar of barley. Hopefully, it will find The Reviewer's approval.

 Figure and Tables: Some figures are too low resolution, please improve it.

Thank you for this comment. We saved all figures as *.pgn files with 600dpi. In this form, we inserted the figures into the manuscript template. The raw immunoblot scan was in a doc file, but the bands are visible in each line, and line 6 shows a strong band of about 15kDa in response to allergenic patient serum.

Reviewer 2 Report

The manuscript identifies a 15KDa  protein from barley responsible for immunogenic properties. This manuscript read well. I have some comments : 

1.      Based on my understanding complete barley genome sequencing is recently done, is it worth to confirm the sequence of B-Fr-VI?

2.      If this is amylase/trypsin inhibitor how closer it is to the published 3-D structure from other families. Having a structure, may be via alpha fold, could be considered for epitope mapping.

3.      Some thermodynamic data is available on similar protein (Scientific Reports volume 8, Article number: 689 (2018)) , looking conserved motifs is another valuable observation to identify immunogenic epitopes thereby identifying peptides.

4.      Minor comment: In methods section all the subsections have extra periods in heading.

Author Response

Dear Reviewer,

Thank you for your favorable review and interest in the topic.

We have corrected the formatting of the text.

Comprehensive coverage of the allergenicity of barley proteins (and other food proteins) in terms of similarity to other species, 3D structure, and the possible presence of conserved proteins is an important part of allergenicity studies of individual food proteins. Such detailed identification of epitopes will also facilitate the search for methods of their inactivation.

Thanks again for your interest in our topic. We add answers to the comments bellow

  1. Based on my understanding, complete barley genome sequencing was recently done. Is it worth confirming the sequence of B-Fr-VI?

Up-to-date barley genome is known. On the EnsemblPants website, it is described

(http://plants.ensembl.org/Hordeum_vulgare/Info/Index#:~:text=Barley%20is%20a%20diploid%20member,diploid%20genomes%20sequenced%20to%20date).

The profile of grain proteins could vary on the species, cultivar, and the conditions to which the grain is subjected after harvesting. We find a 15kDa fraction in crude protein extract from the Barley variety Poldek and the same in local beer. It is almost impossible that the beer has the malt from the same grains we used in experiments. Because of that, we decided that additional sequencing of B-Fr-VI is not necessary for this study. Indeed, the proteomic analysis of the protein profile of the tested 15 kDa fraction would be a "fingerprint" of characterized protein. Still, it would only give information about some identified proteins present in the fraction, not their specific activities. The performed in silico analysis of potential immunogens in this range of molecular mass gave a general insight into a broad spectrum of proteins previously identified in barley from various cultivars and subjected to multiple environmental conditions and parameters of technological processing. Thus, in the authors' opinion, screening in silico was no less valuable than identifying a specific fraction. However, in the future, this fraction will undergo proteomic profiling.

  1. If this is amylase/trypsin inhibitor, how closer it is to the published 3-D structure from other families. Having a structure may be via alpha fold, which could be considered for epitope mapping.

Thank you for the comment.

The reported barley origin amylase/trypsin inhibitor (available and deposited in UniProt and NCBI databases isoforms in the range of 145-171 aa; ID P28041,  P16969, P01086, P32936, P11643) have a high level of homology to the amylase/trypsin inhibitors of other species deposited in allergen databases, e.g., wheat. The homology ranges 82.5% identity (94.4% similarity) E: 1.7e-025 to protein of Triticum aestivum to 90.2% identity (95.8% similarity) E: 6.6e-028 to protein of Triticum turgidum. The in silico analysis of allergenicity was performed with the Allergen Online database tool (last update 14 February 2021) using the Full FASTA 36 search algorithm (E-value Cutoff = 1) and with Allergome database (http://www.allergome.org; last update 10 June 2022) using the NCBI blastp algorithm with E-value Cutoff = 1. According to worldwide accepted algorithms, cross-reactivity is not likely for proteins with less than 50% identity and similarity and E < 1 × 10−4 but for >70% over the entire protein sequence, E < 1 × 10−7 is considered significant. In our case, obtained values confirmed that barley amylase/trypsin inhibitor could likely induce an allergic response. Over 80% of identity and similarity in the 3D model indicates a high similarity of the conformational structure of that proteins. All barley isoforms contain Alpha Fold structures with almost 50% very high and confident values (>70% pLDDT scores). A per-residue confidence score (pLDDT) ranges between 0 and 100. Regions with low pLDDT (low <70) may be unstructured in isolation. However, applied in silico modeling tools use modeling algorithms based on the mapping of sequential but also the conformational level.

In addition, the analyzed fraction VI, in the authors' opinion, could be a subunit of bigger proteins. That is why 23 proteins (mostly 15 kDA ± 10%) and bigger proteins (whose subunits fall within this molecular weight range) were subjected to allergenicity mapping, taking into account the potential sites of conformational epitopes (http://tools.iedb.org/mhcii/).

  1. Some thermodynamic data is available on similar proteins (Scientific Reportsvolume 8, Article number: 689 (2018)), looking conserved motifs is another valuable observation to identify immunogenic epitopes thereby identifying peptides.

Thank you very much for pointing out the work of Kumar A. et al., Docking, thermodynamics and molecular dynamics (MD) studies of a non-canonical protease inhibitor, MP-4, from Mucuna pruriens. Sci Rep. 2018 Jan 12;8(1):689. doi: 10.1038/s41598-017-18733-9.

It was undoubtedly an interesting and valuable manuscript. All thermodynamic studies of proteins are an invaluable source of knowledge, and their detailed analysis should be included in modeling protein allergenicity and monitoring conserved motifs. However, this is a scope of research that requires different specializations and characteristics of the study conducted. In addition, performing such an analysis would require identifying and isolating individual proteins from the fraction (of which there may be even several dozen) and a considerable financial outlay and time. Our goal, however, was to verify the immunogenicity of protein fractions with various molecular weight ranges to determine whether all protein fractions induce the same strong reactions or are more immunogenic, which may translate into further activities in the technology field for obtaining hypoallergenic products.

Nevertheless, we would like to thank you for all the aspects discussed here, thanks to which we could also expand our knowledge.

  1. Minor comment: In methods section all the subsections have extra periods in heading.

Thank you for your comment. We corrected it.

Round 2

Reviewer 1 Report

Accepted suggestion!